# Adaptive End-to-End Metric Learning for Zero-Shot Cross-Domain Slot Filling

**Yuanjun Shi[1], Linzhi Wu[2], Minglai Shao[1*]**

[1]School of New Media and Communication,
Tianjin University, Tianjin, China
[2]School of Life Science and Technology,
University of Electronic Science and Technology of China, Chengdu, China
[1]{switchsyj, shaoml}@tju.edu.cn
[2]lindgew@std.uestc.edu.cn

## Abstract

Recently slot filling has witnessed great development thanks to deep learning and the availability of large-scale annotated data. However, it poses a critical challenge to handle a novel domain whose samples are never seen during training. The recognition performance might be greatly degraded due to severe domain shifts. Most prior works deal with this problem in a two-pass pipeline manner based on metric learning. In practice, these dominant pipeline models may be limited in computational efficiency and generalization capacity because of non-parallel inference and context-free discrete label embeddings. To this end, we re-examine the typical metric-based methods, and propose a new adaptive end-to-end metric learning scheme for the challenging zero-shot slot filling. Considering simplicity, efficiency and generalizability, we present a cascade-style joint learning framework coupled with context-aware soft label representations and slot-level contrastive representation learning to mitigate the data and label shift problems effectively. Extensive experiments on public benchmarks demonstrate the superiority of the proposed approach over a series of competitive baselines.[1]

## 1 Introduction

Slot filling, as an essential component widely exploited in task-oriented conversational systems, has attracted increasing attention recently (Zhang and Wang, 2016; Goo et al., 2018; Gangadharaiah and Narayanaswamy, 2019). It aims to identify a specific type (*e.g.*, `artist` and `playlist`) for each slot entity from a given user utterance. Owing to the rapid development of deep neural networks and with help from large-scale annotated data, research on slot filling has made great progress with considerable performance improvement (Qin et al., 2019; Wu et al., 2020; Qin et al., 2020, 2021).

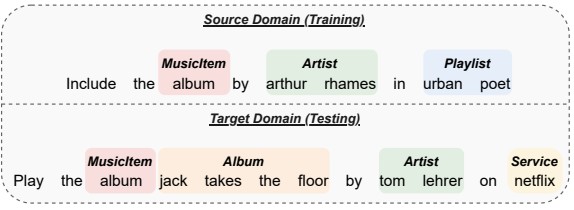

Figure 1: Examples from SNIPS dataset. Apart from data distribution shifts, the target domain contains novel slot types that are unseen in the source domain label space (*e.g.*, *Album* and *Service*). Moreover, the slot entities tend to embody domain-specific nature in contrast to the counterpart contexts.

Despite the remarkable accomplishments, there are at least two potential challenges in realistic application scenarios. First is the data scarcity problem in specific target domains (*e.g.*, *Healthcare* and *E-commerce*). The manually-annotated training data in these domains is probably unavailable, and even the unlabeled training data might be hard to acquire (Jia et al., 2019; Liu et al., 2020a). As a result, the performance of slot filling models may drop significantly due to extreme data distribution shifts. The second is the existence of label shifts (as shown in the example in Figure 1). The target domain may contain novel slot types unseen in the source-domain label space (Liu et al., 2018; Shah et al., 2019; Liu et al., 2020b; Wang et al., 2021), namely there is a mismatch between different domain label sets. This makes it difficult to apply the source models to completely unseen target domains that are unobservable during the training process.

Zero-shot domain generalization has been shown to be a feasible solution to bridge the gap of domain shifts with no access to data from the target domain. Recent dominating advances focus on the two-step pipeline fashion to learn the zero-shot model using the metric learning paradigms (Shah et al., 2019; Liu et al., 2020b; He et al., 2020; Wang et al., 2021; Siddique et al., 2021). Nevertheless, besides inefficient inference resulted from non-parallelization,

---

*Corresponding author.

[1]The source code is available at https://github.com/Switchsyj/AdaE2ML-XSF.

the generalization capability of these models may be limited due to lack of knowledge sharing between sub-modules, and context-independent discrete static label embeddings. Although the alternative question-answering (QA) based methods (Du et al., 2021; Yu et al., 2021; Liu et al., 2022) are able to achieve impressive results, they need to manually design and construct the questions/queries, essentially introducing detailed descriptive information about the slot types.

In this work, we revisit the metric-based zero-shot cross-domain slot filling under challenging domain (both data and label) shifts. We propose an adaptive end-to-end metric learning scheme to improve the efficiency and effectiveness of the zero-shot model in favor of practical applications. For one thing, we provide a cascade-style joint learning architecture well coupled with the slot boundary module and type matching module, allowing for knowledge sharing among the sub-modules and higher computational efficiency. Moreover, the soft label embeddings are adaptively learnt by capturing the correlation between slot labels and utterance. For another, since slot terms with same types tend to have the semantically similar contexts, we propose a slot-level contrastive learning scheme to enhance the slot discriminative representations within different domain context. Finally, to verify the effectiveness of the proposed method, we carry out extensive experiments on different benchmark datasets. The empirical studies show the superiority of our method, which achieves effective performance gains compared to several competitive baseline methods.

Overall, the main contributions can be summarized as follows: (1) Compared with existing metric-based methods, we propose a more efficient and effective end-to-end scheme for zero-shot slot filling, and show our soft label embeddings perform much better than previous commonly-used static label representations. (2) We investigate the slot-level contrastive learning to effectively improve generalization capacity for zero-shot slot filling. (3) By extensive experiments, we demonstrate the benefits of our model in comparison to the existing metric-based methods, and provide an insightful quantitative and qualitative analysis.

## 2 Methodology

In this section, we first declare the problem to be addressed about zero-shot slot filling, and then elaborate our solution to this problem.

### 2.1 Problem Statement

Suppose we have the source domain $\mathcal{D}_\mathcal{S} = \{(\mathbf{x}_i^\mathcal{S}, \mathbf{y}_i^\mathcal{S})\}_{i=1}^{N_\mathcal{S}}$ with $N_\mathcal{S}$ labeled samples from distribution $P^\mathcal{S}$, and the (testing) target domain $\mathcal{D}_\mathcal{T} = \{(y_j^\mathcal{T})\}_{j=1}^{C}$ with $C$ slot types from target distribution $P^\mathcal{T}$. We define $\Omega_\mathcal{S}$ as the label set of source domain $\mathcal{D}_\mathcal{S}$, and $\Omega_\mathcal{T}$ as the label set of target domain $\mathcal{D}_\mathcal{T}$. $\Omega_{sh} = \Omega_\mathcal{S} \cap \Omega_\mathcal{T}$ denotes the common slot label set shared by $\mathcal{D}_\mathcal{S}$ and $\mathcal{D}_\mathcal{T}$. In the zero-shot scenario, the label sets between different domains may be mismatching, thus $\Omega_{sh} \subseteq \Omega_\mathcal{S}$ and $P^\mathcal{S} \neq P^\mathcal{T}$. The goal is to learn a robust and generalizable zero-shot slot filling model that can be well adapted to novel domains with unknown testing distributions.

### 2.2 Overall Framework

In order to deal with variable slot types within an unknown domain, we discard the standard sequence labeling paradigm by cross-labeling (*e.g.*, B-playlist, I-playlist). Instead, we adopt a cascade-style architecture coupled with the slot boundary module and typing module under a joint learning framework. The boundary module is used to detect whether the tokens in an utterance are slot terms or not by the CRF-based labeling method with BIO schema, while the typing module is used to match the most likely type for the corresponding slot term using the metric-based method. Since pre-training model is beneficial to learn general representations, we adopt the pre-trained BERT (Devlin et al., 2019) as our backbone encoder[2]. Figure 2 shows the overall framework, which is composed of several key components as follows:

**Context-aware Label Embedding** Let $\mathbf{c} = [c_1, \cdots, c_{|\Omega_\mathcal{S}|}]$ $(c_i \in \Omega_\mathcal{S})$ denotes a slot label sequence consisting of all the elements of $\Omega_\mathcal{S}$. Given an input utterance sequence $\mathbf{x} = [x_1, \cdots, x_n]$ of $n$ tokens with the corresponding ground-truth boundary label sequence $\mathbf{y}^{bd} = [y_1^{bd}, \cdots, y_n^{bd}]$ $(y_i^{bd} \in \{\text{B}, \text{I}, \text{O}\})$ and slot label sequence $\mathbf{y}^{sl} = [y_1^{sl}, \cdots, y_n^{sl}]$ $(y_i^{sl} \in \Omega_\mathcal{S})$, the slot label sequence acts as a prefix of the input utter-

---

[2]Notice that we assume the BERT model is used as our encoder, but our method can also be integrated with other model architectures (*e.g.*, RoBERTa (Liu et al., 2019b)).

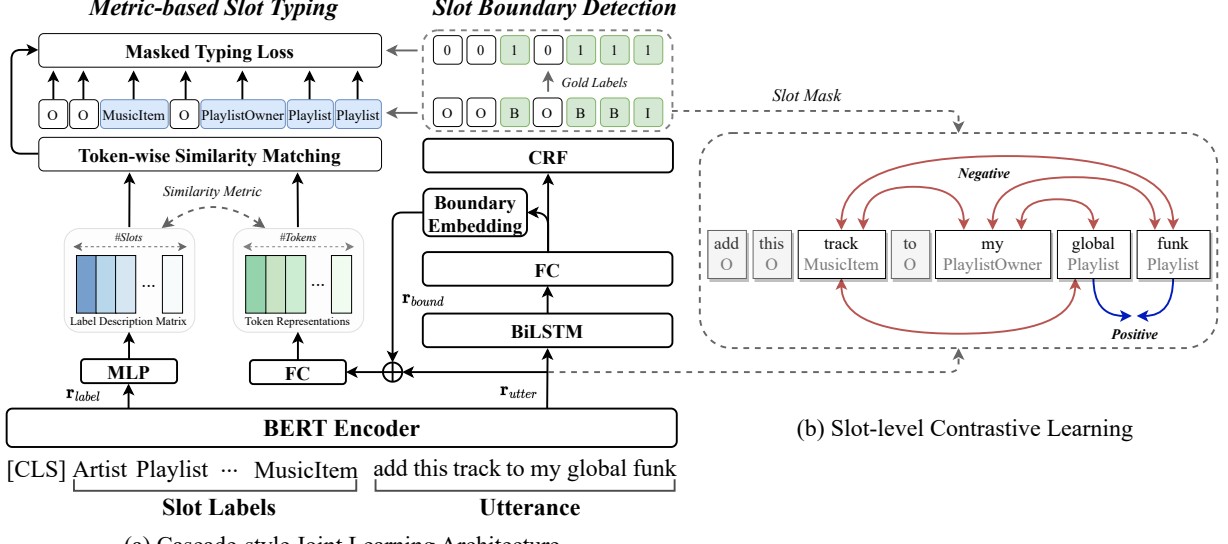

(a) Cascade-style Joint Learning Architecture

(b) Slot-level Contrastive Learning

Figure 2: Illustration of the overall framework. Figure (a) shows the cascade-style joint learning architecture coupled with two core components: **Metric-based Slot Typing** and **Slot Boundary Detection**. Figure (b) shows the slot-level contrastive learning module used only during training. The slot entity tokens with the same type are positive pairs (*i.e.* the blue lines) while those with different types are negative ones (*i.e.* the red lines).

ance, which is then encoded by BERT[3]:

$$[\mathbf{r}_{label}; \mathbf{r}_{utter}] = \text{BERT}([\mathbf{c}; \mathbf{x}]), \quad (1)$$

where $\mathbf{r}_{label}$ and $\mathbf{r}_{utter}$ denote the fused contextual representations of the label and utterance sequence, respectively.

For each slot type, the slot label matrix is obtained by averaging over the representations of the slot label tokens. Unlike the conventional discrete and static label embeddings (Liu et al., 2020b; Siddique et al., 2021; Ma et al., 2022) that capture the semantics of each textual label separately, we attempt to build the label-utterance correlation, and the adaptive interaction between the slot labels and utterance tokens encourages the model to learn the context-aware soft label embeddings dynamically, which will be exploited as the supervision information for the metric learning.

**Slot Boundary Detection**  To determine the slot terms, we obtain the contextualized latent representations of the utterance through a single-layer BiLSTM,

$$\mathbf{h}_{utter} = \text{BiLSTM}(\mathbf{r}_{utter}). \quad (2)$$

Then, a CRF layer is applied to the slot boundary decoding, aiming to model the boundary label de-

pendency. The negative log-likelihood objective function can be formulated as follows:

$$\mathbf{e} = \text{Linear}(\mathbf{h}_{utter}),$$
$$score(\mathbf{x}, \mathbf{y}) = \sum_{i=1}^{n}(\mathbf{T}_{\mathbf{y}_{i-1}, \mathbf{y}_i} + \mathbf{e}_i[\mathbf{y}_i]),$$
$$\mathcal{L}_{bdy} = -\log p(\mathbf{y}^{bd}|\mathbf{x}) \quad (3)$$
$$= -\log \frac{\exp(score(\mathbf{x}, \mathbf{y}^{bd}))}{\sum_{\mathbf{y}' \in \mathcal{Y}_{\mathbf{x}}} \exp(score(\mathbf{x}, \mathbf{y}'))},$$

where $\mathbf{e} \in \mathbb{R}^{n \times 3}$ denotes the three-way emission vectors containing boundary information, $\mathbf{T}$ is the $3 \times 3$ learnable label transition matrix, and $\mathcal{Y}_{\mathbf{x}}$ is the set of all possible boundary label sequences of utterance $\mathbf{x}$. While inference, we employ the Viterbi algorithm (Viterbi, 1967) to find the best boundary label sequence.

**Metric-based Slot Typing**  Although slot boundary module can select the slot terms from an utterance, it fails to learn discriminative slot entities. Thus, we design a typing module to achieve it in parallel by semantic similarity matching between slot labels and utterance tokens.

Concretely, we take advantage of the above boundary information to locate the slot entity tokens of the utterance. We specially exploit the soft-weighting boundary embedding vectors for enabling differentiable joint training, which are combined with the contextual utterance representations

---
[3]Considering the slot label sequence is not a natural language sentence linguistically, we remove the [SEP] token used to concatenate sentence pairs in BERT.

to obtain the boundary-enhanced representations:

$$\mathbf{r}_{bound} = \text{softmax}(\mathbf{e}) \cdot \mathbf{E}_b,$$
$$\mathbf{u} = \text{Linear}(\text{Concat}(\mathbf{r}_{utter}, \mathbf{r}_{bound})), \quad (4)$$

where $\mathbf{E}_b \in \mathbb{R}^{3 \times d_b}$ is a look-up table to store trainable boundary embeddings, and $d_b$ indicates the embedding dimension. Meanwhile, the label embeddings are calculated by a bottleneck module consisting of an up-projection layer and a down-projection layer with a GELU (Hendrycks and Gimpel, 2016) nonlinearity:

$$\mathbf{v} = \text{Linear}_{up}(\text{GELU}(\text{Linear}_{dw}(\mathbf{r}_{label}))). \quad (5)$$

Furthermore, we leverage token-wise similarity matching between L2-normalized utterance representations and label embeddings. Since the slot entities are our major concern for predicting types, we ignore the non-entity tokens by mask provided by the boundary gold labels, resulting in the slot typing loss function defined as follows:

$$\mathcal{L}_{typ} = -\sum_{i=1}^{n} \mathbb{1}_{[y_i^{bd} \neq 0]} \log \frac{\exp(\langle \mathbf{u}_i, \text{sg}(\mathbf{v}_{i^*}) \rangle)}{\sum_{j=1}^{|\Omega_{\mathcal{S}}|} \exp(\langle \mathbf{u}_i, \text{sg}(\mathbf{v}_j) \rangle)}$$
$$-\sum_{i=1}^{n} \mathbb{1}_{[y_i^{bd} \neq 0]} \log \frac{\exp(\langle \text{sg}(\mathbf{u}_i), \mathbf{v}_{i^*} \rangle)}{\sum_{j=1}^{|\Omega_{\mathcal{S}}|} \exp(\langle \text{sg}(\mathbf{u}_i), \mathbf{v}_j \rangle)}, \quad (6)$$

where $\langle \cdot, \cdot \rangle$ measures the cosine similarity of two embeddings, $\text{sg}(\cdot)$ stands for the stop-gradient operation that does not affect the forward computation, $i^*$ indicates the index corresponding to the gold slot label $y_i^{sl}$, and $\mathbb{1}_{[y_i^{bd} \neq 0]} \in \{0, 1\}$ is an indicator function, evaluating to 1 if $y_i^{bd}$ is a non-0 tag. Eq. 6 makes sure the label embeddings act as the supervision information (the *1st* term) and meanwhile are progressively updated (the *2nd* term).

**Slot-level Contrastive Learning**   Recent line of works have investigated the instance-level contrastive learning by template regularization (Shah et al., 2019; Liu et al., 2020b; He et al., 2020; Wang et al., 2021). As slots with the same types tend to have the semantically similar contexts, inspired by Das et al. (2022), we propose to use the slot-level contrastive learning to facilitate the discriminative slot representations that may contribute to adaptation robustness.[4]

---

[4]Different from Das et al. (2022), we do not use the Gaussian embeddings produced by learnt Gaussian distribution parameters. There are two main reasons: one is to ensure the stable convergence of training, and the other is that the token representations may not follow normal distribution.

More specifically, we define a supervised contrastive objective by decreasing the similarities between different types of slot entities while increasing the similarities between the same ones. We just pay attention to the slot entities by masking out the parts with 0 boundary labels. Then, we gather *in-batch* positive pairs $\mathcal{P}^+$ with the same slot type and negative pairs $\mathcal{P}^-$ with different ones:

$$\mathbf{s} = \text{ReLU}(\text{Linear}(\mathbf{r}_{utter})),$$
$$\mathcal{P}^+ = \{(\mathbf{s}^i, \mathbf{s}^j) | y_i^{sl} = y_j^{sl}, i \neq j\}, \quad (7)$$
$$\mathcal{P}^- = \{(\mathbf{s}^i, \mathbf{s}^j) | y_i^{sl} \neq y_j^{sl}, i \neq j\},$$

where $\mathbf{s}$ denotes the projected point embeddings, and all example pairs are extracted from a mini-batch. Furthermore, we adapt the NT-Xent loss (Chen et al., 2020) to achieve the slot-level discrimination, and the contrastive learning loss function can be formulated as:

$$\mathcal{L}_{ctr} = -\log \frac{\frac{1}{|\mathcal{P}^+|} \sum_{(\mathbf{s}^i, \mathbf{s}^j) \in \mathcal{P}^+} \exp(d(\mathbf{s}^i, \mathbf{s}^j)/\tau)}{\sum_{(\mathbf{s}^i, \mathbf{s}^j) \in \mathcal{P}} \exp(d(\mathbf{s}^i, \mathbf{s}^j)/\tau)}, \quad (8)$$

where $\mathcal{P}$ denotes $\mathcal{P}^+ \cup \mathcal{P}^-$, $d(\cdot, \cdot)$ denotes the distance metric function (*e.g.*, cosine similarity distance), and $\tau$ is a temperature parameter. We will investigate different kinds of metric functions in the following experiment section.

### 2.3 Training and Inference

During training, our overall framework is optimized end-to-end with min-batch. The final training objective is to minimize the sum of the all loss functions:

$$\mathcal{L} = \mathcal{L}_{bdy} + \mathcal{L}_{typ} + \mathcal{L}_{ctr}, \quad (9)$$

where each part has been defined in the previous subsections. During inference, we have the slot type set of the target domain samples, and the testing slot labels constitute the label sequence, which is then concatenated with the utterance as the model input. The CRF decoder predicts the slot boundaries of the utterance, and the predicted slot type corresponds to the type with the highest-matching score. We take the non-0-labeled tokens as slot terms while the 0-labeled tokens as the context.

## 3 Experiments

### 3.1 Datasets and Settings

To evaluate the proposed method, we conduct the experiments on the SNIPS dataset for zero-

| Domain Model↓ (Src→Tgt, Unseen Rate) → | ATP (48→5, 40%) | BR (39→14, 57%) | GW (44→9, 44%) | PM (44→9, 55%) | RB (46→7, 71%) | SCW (52→2, 0%) | SSE (46→7, 57%) | Avg. |
|---|---|---|---|---|---|---|---|---|
| Coach_BERT (Liu et al., 2020b) | 50.28 | 31.87 | 52.30 | 31.75 | 23.33 | 70.76 | 29.33 | 41.37 |
| PCLC_BERT (Wang et al., 2021) | 30.38 | 20.89 | 32.99 | 25.55 | 20.76 | 62.40 | 13.82 | 29.54 |
| LEONA_BERT (Siddique et al., 2021) | 51.23 | 46.68 | 68.72 | 43.20 | 25.23 | 47.01 | 27.99 | 44.01 |
| QASF_BERT † (Du et al., 2021) | 59.29 | 43.13 | 59.02 | 33.62 | 33.34 | 59.90 | 22.83 | 44.45 |
| SLMRC† (Liu et al., 2022) | 63.21 | 60.11 | 65.23 | 50.16 | 32.78 | 55.17 | 30.67 | 51.77 |
| GZPL_{T5−Large}‡ (Li et al., 2023) | 59.83 | 61.23 | 62.58 | 62.73 | 45.88 | 71.30 | 48.26 | 58.82 |
| Ours (w/o Slot-CL) | 61.13 | 41.67 | 71.47 | 34.77 | 30.75 | 68.81 | 34.64 | 49.03 |
| Ours | 61.13 | 42.35 | 69.87 | 36.24 | 33.25 | 70.81 | 34.06 | **49.67** |

Table 1: F1-scores of zero-shot slot filling across different domains. Slot-CL denotes the slot-level contrastive learning. We show the number of labels in the source and target domains. The unseen rate refers to the proportion of non-overlapped source-target domain labels in the target label set. † denotes the QA-based methods that introduce manually-designed query for each slot label. ‡ denotes the generative method along with prompts for each slot label.

shot settings (Coucke et al., 2018), which contains 39 slot types across seven different domains: AddToPlaylist (ATP), BookRestaurant (BR), GetWeather (GW), PlayMusic (PM), RateBook (RB), SearchCreativeWork (SCW) and SearchScreeningEvent (SSE). Following previous studies (Liu et al., 2020b; Siddique et al., 2021), we choose one of these domains as the target domain never used for training, and the remaining six domains are combined to form the source domain. Then, we split 500 samples in the target domain as the development set and the remainder are used for the test set. Moreover, we consider the case where the label space of the source and target domains are exactly the same, namely the zero-resource setting (Liu et al., 2020a) based on named entity recognition (NER) task. We train our model on the CoNLL-2003 (Sang and Meulder, 2003) dataset and evaluate on the CBS SciTech News dataset (Jia et al., 2019).

## 3.2 Baselines

We compare our method with the following competitive baselines using the pre-trained BERT as encoder: (1) **Coach**. Liu et al. (2020b) propose a two-step pipeline matching framework assisted by template regularization; (2) **PCLC**. Wang et al. (2021) propose a prototypical contrastive learning method with label confusion; (3) **LEONA**. Siddique et al. (2021) propose to integrate linguistic knowledge (e.g., external NER and POS-tagging cues) into the basic framework.

Although not our focused baselines, we also compare against the advanced generative baselines (Li et al., 2023) with T5-Large and QA-based methods (Du et al., 2021; Liu et al., 2022) that require manual efforts to convert slot type descriptions into sentential queries/questions, and process by means of the machine reading comprehension (MRC) ar-

chitecture (Li et al., 2020).

## 3.3 Implementation Details

We use the pre-trained uncased BERT_BASE model[5] as the backbone encoder. The dimension of the boundary embedding is set to 10. We use 0.1 dropout ratio for slot filling and 0.5 for NER. For the contrastive learning module, we use the cosine metric function and select the optimal temperature $\tau$ from 0.1 to 1. During training, the AdamW (Loshchilov and Hutter, 2019) optimizer with a mini-batch size 32 is applied to update all trainable parameters, and the initial learning rate is set to 2e-5 for BERT and 1e-3 for other modules. All the models are trained on NIVIDIA GeForce RTX 3090Ti GPUs for up to 30 epochs. The averaged F1-score over five runs is used to evaluate the performance. The best-performing model on the development set is used for testing.

## 3.4 Zero-Shot Slot Filling

As shown in Table 1, our method achieves more promising performance than previously proposed metric-based methods on various target domains, with an average about 5% improvements compared with the strong baseline LEONA. We attribute it to the fact that our proposed joint learning model make full use of the sub-modules, and the context-aware soft label embeddings provide better prototype representations. Moreover, we also observe that the slot-level contrastive learning plays an important role in improving adaptation performance. Our model with *Slot-CL* obtains consistent performance gains over almost all the target domains except for the SSE domain. We suspect that it may result from slot entity confusion. For example, for slot entities "*cinema*" and

---

[5] https://huggingface.co/bert-base-uncased

"*theatre*" from SSE, they are usually annotated with `object_location_type`, but "*cinemas*" in "*caribbean cinemas*" and "*theatres*" in "*star theatres*" are annotated with `location_name`, which is prone to be misled by the contrastive objective. Additionally, without introducing extra manual prior knowledge, our method achieves very competitive performance compared with the QA-based baselines.

### 3.5 Zero-Resource NER

In particular, we examine our method in the zero-resource NER setting. As presented in Table 2, our method is also adaptable to this scenario, and exceed or match the performance of previous competitive baselines. Meanwhile, the slot-level contrastive learning can yield effective performance improvements.

| Model | F1 |
|---|---|
| Liu et al. (2020a) | 69.53 |
| Jia et al. (2019) | 73.59 |
| Devlin et al. (2019) | 74.23 |
| Jia and Zhang (2020) | 75.19 |
| Wu et al. (2022) | 75.06 |
| Ours (w/o Slot-CL) | 74.41 |
| Ours | **75.29** |

Table 2: NER results on the target domain (i.e., SciTech News).

### 3.6 Ablation Study and Analysis

In order to better understand our method, we further present some quantitative and qualitative analyses that provides some insights into why our method works and where future work could potentially improve it.

**Inference Speed** One advantage of our framework is the efficient inference process benefiting from the well-parallelized design. We evaluate the speed by running the model one epoch on the BookRestaurant test data with batch size set to 32. Results in Table 3 show that our method achieves $\times 13.89$ and $\times 7.06$ speedup compared with the advanced metric-based method (i.e., LEONA) and QA-based method (i.e., SLMRC). This could be attributed to our batch-wise decoding in parallel. On the one hand, previous metric-based methods use the two-pass pipeline decoding process and instance-wise slot type prediction. On the other

hand, the QA-based methods require introducing different queries for all candidate slot labels regarding each utterance, increasing the decoding latency of a single example.

| Model | Time Cost (s) | Speedup |
|---|---|---|
| Coach$_{BERT}$ | 70.98 | 17.14$\times$ |
| PCLC$_{BERT}$ | 76.61 | 18.50$\times$ |
| LEONA | 57.49 | 13.89$\times$ |
| SLMRC | 29.21 | 7.06$\times$ |
| Ours | 4.14 | 1.00$\times$ |

Table 3: Comparison of inference efficiency. Speedup denotes the ratio of time taken by slot prediction part of different models to run one epoch on the BookRestaurant with batch size 32.

**Label-Utterance Interaction** Here we examine how our model benefits from the label-utterance interaction. As presented in Table 4, the performance of our model drops significantly when eliminating the interaction from different aspects , justifying our design. Compared to the other degraded interaction strategies, the utterance-label interaction helps learn the context-aware label embeddings, namely the utterance provides the context cues for the slot labels. Furthermore, we also notice that interaction between slot labels also makes sense. When only let each slot label attend to itself and the utterance, we observe the performance drop probably due to the loss of discriminative information among different slot labels.

| Interaction Strategy | F1 |
|---|---|
| Ours (w/o Slot-CL) | **49.03** |
| w/o Label $\rightarrow$ Utterance | 45.42 |
| w/o Utterance $\rightarrow$ Label | 45.24 |
| w/o Label $\leftrightarrow$ Utterance | 47.25 |
| w/o Label $\leftrightarrow$ Label | 45.24 |

Table 4: Comparisons of different label-utterance interaction strategies for slot filling.

**Effect of Context-aware Label Embedding** We study the effect of different types of label embeddings. Figure 3 shows the comparison results. We can see that the proposed context-aware soft label embedding outperforms other purely discrete or decoupled embeddings, including discrete BERT, decoupled BERT or GloVe (Pennington et al.,

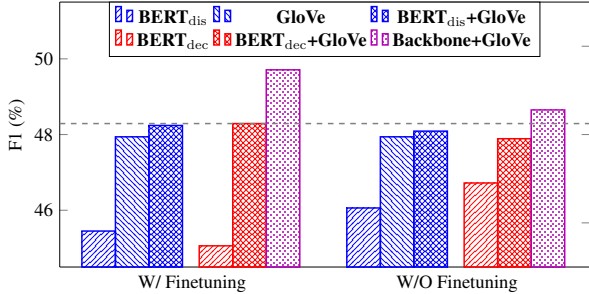

Figure 3: Comparisons of different label representations for slot filling. BERT$_{dis}$ and BERT$_{dec}$ denote the discrete label embedding and the label-utterance decoupled embedding, respectively. The gray dashed line represents the performance of our model without the slot-level contrastive learning.

2014) embeddings. Interestingly, when fine-tuning, we find that BERT$_{dis}$ works slightly better than BERT$_{dec}$, as it might be harmful to tune soft label embeddings without utterance contexts. Furthermore, we observe a significant improvement of our model when incorporating the GloVe static vectors, suggesting that richer label semantics can make a positive difference. Meanwhile, the discrete or decoupled label embeddings without fine-tuning may yield better results.

**Metric Loss for Contrastive Learning**  Here we explore several typical distance metric functions (including Cosine, MSE, Smooth L1, and KL-divergence) for the slot-level contrastive objective, and we also consider the influence of temperature $\tau$. Figure 4 reveals that the temperature value directly affects the final performance. Also, it shows better results overall at around $\tau = 0.5$ for each metric function we take. We select the cosine similarity function as our desired distance metric function, due to its relatively good performance.

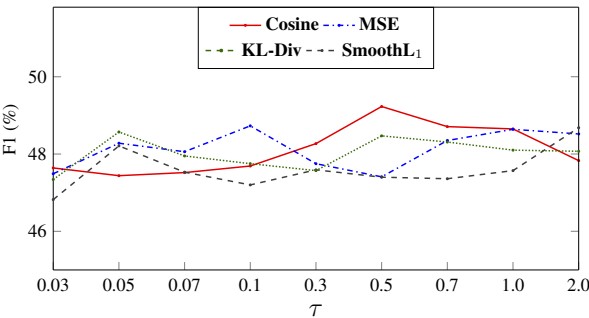

Figure 4: Comparisons of different metric functions with varying temperature $\tau$ for slot filling.

**Few-Shot Setting**  To verify the effectiveness of our method in the few-shot setting where the target domain has a small amount of training examples, we conduct experiments in the 20-shot and 50-shot scenarios. In line with previous works, we take the first $K$ examples in the development set for training named the $K$-Shot scenario and the remaining keeps for evaluation.

Table 5 illustrates that our method achieves superior performance compared with other representative metric-based methods. However, we also notice that our method without the slot-level contrastive learning obtains limited absolute improvements as data size increase, indicating the slot-level contrastive learning performs better in this case.

| Model | 20-Shot (1%) | 50-Shot (2.5%) |
|---|---|---|
| Coach | 64.27 | 75.51 |
| LEONA | 71.10 | 76.42 |
| PCLC | 54.32 | 72.18 |
| Ours | **75.16** | **82.39** |
| w/o Slot-CL | 70.49 | 81.42 |

Table 5: F1-scores over all target domains in the 20-shot and 50-shot settings.

**Unseen Slot Generalization**  Since label shift is a critical challenge in zero-shot learning, to verify the generalization capacity, we specifically test our method on the unseen target data. Following Liu et al. (2022), we split the dataset into the seen and unseen group, where we only evaluate on unseen slot entities during training in the unseen$_{slot}$ group, while evaluate on the whole utterance in the unseen$_{uttr}$ group. From Table 6, our method performs better than other metric-based baselines, showing the superiority of our method for unseen domain generalization.

| Model | Seen | Unseen$_{uttr}$ | Unseen$_{slot}$ |
|---|---|---|---|
| Coach | 51.73 | 34.23 | 11.66 |
| PCLC | 58.76 | 35.08 | 10.92 |
| LEONA | 63.54 | 40.06 | 12.32 |
| Ours | **65.80** | **46.85** | **21.23** |
| w/o Slot-CL | 64.24 | 41.04 | 14.57 |

Table 6: F1-scores for seen and unseen slots over all the target domains.

**Cross-Dataset Setting** Considering slot labels and utterances may vary significantly across different datasets, we further evaluate the proposed method under the cross-dataset scenario, a more challenging setting. Here we introduce another popular slot filling dataset ATIS (Liu et al., 2019a). It is used for the target (source) domain data while the SNIPS for the source (target) domain data[6], as shown in Table 7. The results confirm that our method still works well in this challenging setting.

| Src→Tgt | SNIPS→ATIS | ATIS→SNIPS |
|---------|-----------|-----------|
| Coach | 14.35 | 9.76 |
| LEONA | 20.80 | 14.36 |
| Ours | 27.01 | 16.61 |

Table 7: F1-scores in the cross-dataset setting.

**Visualization** Figure 5 shows the visualization of normalized slot entity representations before similarity matching using t-SNE dimensionality reduction algorithm (van der Maaten and Hinton, 2008). Obviously, our method can better obtain well-gathered clusters when introducing the slot-level contrastive learning, facilitating the discriminative entity representations.

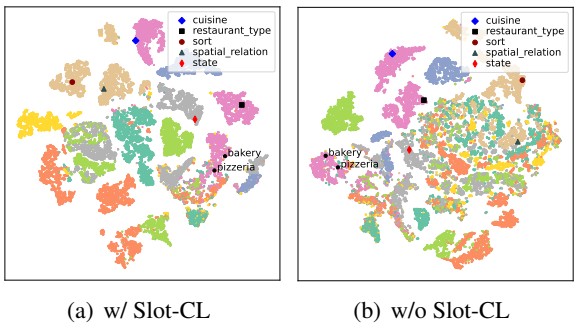

   (a) w/ Slot-CL       (b) w/o Slot-CL

Figure 5: t-SNE visualization of the normalized representations of different slot entities drawn from the BookRestaurant that contains much unseen slot labels.

## 4 Related Work

**Zero-shot Slot Filling** In recent years, zero-shot slot filling has received increasing attention. A dominating line of research is the metric-learning method, where the core idea is to learn a prototype representation for each category and classify

test data based on their similarities with prototypes (Snell et al., 2017). For slot filling, the semantic embeddings of textual slot descriptions usually serve as the prototype representations (Bapna et al., 2017; Lee and Jha, 2019; Zhu et al., 2020). Shah et al. (2019) utilize both the slot description and a few examples of slot values to learn semantic representations of slots. Furthermore, various two-pass pipeline schemes are proposed by separating the slot filling task into two steps along with template regularization (Liu et al., 2020b), adversarial training (He et al., 2020), contrastive learning (Wang et al., 2021), linguistic prior knowledge (Siddique et al., 2021). However, these mostly utilize the context-free discrete label embeddings, and the two-pass fashion has potential limitations due to a lack of knowledge sharing between sub-modules as well as inefficient inference. These motivate us to exploit the context-aware label representations under an end-to-end joint learning framework.

Another line of research is the QA-based methods that borrow from question-answering systems, relying on manually well-designed queries. Du et al. (2021) use a set of slot-to-question generation strategies and pre-train on numerous synthetic QA pairs. Yu et al. (2021) and Liu et al. (2022) apply the MRC framework (Li et al., 2020) to overcome the domain shift problem. Heo et al. (2022) modify the MRC framework into sequence-labeling style by using each slot label as query. Li et al. (2023) introduce a generative framework using each slot label as prompt. In our work, we mainly focus on the metric-based method without intentionally introducing external knowledge with manual efforts.

**Contrastive Learning** The key idea is to learn discriminative feature representations by contrasting positive pairs against negative pairs. Namely, those with similar semantic meanings are pushed towards each other in the embedding space while those with different semantic meanings are pulled apart each other. Yan et al. (2021) and Gao et al. (2021) explore instance-level self-supervised contrastive learning where sample pairs are constructed by data augmentation. Khosla et al. (2020) further explore the supervised setting by contrasting the set of all instances from the same class against those from the other classes. Das et al. (2022) present a token-level supervised contrastive learning solution to deal with the few-shot NER task by means of Gaussian embeddings.

Previous studies for slot filling mainly focus on

---

[6]We ignore the evaluation on the SGD (Rastogi et al., 2020), which is a fairly large-scale dataset with extremely unbalanced label distributions.

instance-level contrastive learning, which may be sub-optimal for a fine-grained sequence labeling task. Inspired by supervised contrastive learning, we leverage a slot-level contrastive learning scheme for zero-shot slot filling to learn the discriminative representations for domain adaptation. For all existing slot entities within a mini-batch, we regard those with the same type as the positive example pairs and those with different type as negative ones.

## 5 Conclusion

In this paper, we tackle the problem of generalized zero-shot slot filling by the proposed end-to-end metric learning based scheme. We propose a cascade-style multi-task learning framework to efficiently detect the slot entity from a target domain utterance. The context-aware soft label embeddings are shown to be superior to the widely-used discrete ones. Regarding domain adaptation robustness, we propose a slot level contrastive learning scheme to facilitate the discriminative representations of slot entities. Extensive experiments across various domain datasets demonstrate the effectiveness of the proposed approach when handling unseen target domains. Our investigation also confirms that semantically richer label representations enable help further boost the recognition performance, which motivates us to further explore external knowledge enhanced soft label embeddings for advancing the metric-based method.

## Limitations

Although our work makes a further progress in the challenging zero-shot slot filling, it is subject to several potential limitations. Firstly, since slot label sequence is used as the prefix of the utterance, this directly results in a long input sequence. Secondly, our method may be negatively affected by severe label ambiguity. There are some slot entities with rather similar semantics, leading to wrong slot type predictions. For example, "*book a manadonese restaurant*", the slot entity type of "*manadonese*" is actually `cuisine`, but is easily identified as `country`. One major reason is that some utterances are relatively short and lack sufficient contextual cues. Thirdly, the recognition performance of metric-based methods may remain difficult to exceed that of advanced QA-based or generative methods due to the fact that the latter manually introduces detailed slot label description by well-designed queries or prompts.

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

# A   Appendix

## A.1   Code Implementation

Here we present the core pseudo code of the proposed method.

```python
class End2endSLUTagger(nn.Module):
    def forward(bert_inp, num_type,
    lbl_bd, lbl_typ):
        # (bsz, seq_len, hsz)
        bert_repr = bert(*bert_inp)
        r_utter = bert_repr[:,num_type
    +1:]
        r_label = bert_repr[:,1:num_type
    +1]

        # label adapter in Eq.5
        v = adapter(r_label) + r_label
        h_utter = lstm(r_utter)
        e = proj(h_utter)
        r_bound = matmul(softmax(e),
    bound_emb.weight)
        # (bsz, 1)
        l_bdy = crf(e, lbl_bd)
        u = proj(concat(token_repr,
    r_bound))
```

```python
        # (bsz, seq_len)
        lbl_score = matmul(normalize(u),
    normalize(v).T.detach())
        l_sglbl = cross_entropy(sg_score
    , lbl_typ)
        l_sglbl *= lbl_bd.ne('O')

        utt_score = matmul(normalize(u).
    detach(), normalize(v).T)
        l_sgutt = cross_entropy(
    utt_score, lbl_typ)
        l_sgutt *= lbl_bd.ne('O')
        # (bsz, 1)
        l_typ = l_sglbl.sum(-1) +
    l_sgutt.sum(-1)
        # filter out paddings and non-
    slot tokens
        f_utt = proj(r_utter)
        filt_ids = lbl_typ != idx_O
        filt_emb = f_utt[filt_ids]
        filt_typ = lbl_typ[filt_ids]
        # repeat on the second dimension
        inter_emb = filt_emb.unsqueeze
    (1).repeat(1, num_slot, 1).view(
    num_slot*num_slot, -1)
        int_typ = filt_typ.unsqueeze(1).
    repeat(1, num_slot).view(num_slot*
    num_slot)
        # repeat on the first dimension
        rept_emb = filt_emb.unsqueeze(0)
    .repeat(num_slot, 1, 1).view(
    num_slot*num_slot, -1)
        rept_typ = filt_typ.unsqueeze(0)
    .repeat(num_slot, 1).view(num_slot*
    num_slot)
        sim_score = cosine_similarity(
    inter_emb, rept_emb)
        # view as (num_slot, num_slot)
        denom_mask = (inter_emb !=
    rept_emb)
        numer_mask = denom * (int_typ ==
     rept_typ)
        loss = softmax(sim_score) /
    temperature
        # # calculate Slot-CL with Eq.8,
     (bsz, 1)
        l_ctr = -(loss*numer_mask).log()
     + (loss*denom_mask).log() +
    num_mask.sum().log()
        l_ctr = l_ctr.mean()

        return l_bdy + l_typ + l_ctr
```

Listing 1: Pseudo code for our proposed method.