# OpenReview forum: "Adaptive End-to-End Metric Learning for Zero-Shot Cross-Domain Slot Filling"
_EMNLP/2023/Conference — EMNLP 2023 Main_

### Official Review · Reviewer_yN6g · 2023-07-20

**Soundness:** 3

**Excitement:**

4: Strong: This paper deepens the understanding of some phenomenon or lowers the barriers to an existing research direction.

**Paper Topic And Main Contributions:**

This paper re-examines the typical metric-based methods, and proposes a new adaptive end-to-end metric learning scheme for the challenging zero-shot slot filling. And the experimental results also demonstrate the effectiveness of our method.

**Reasons To Accept:**

(1) This paper is well-written and the proposed idea is very interesting!
(2) Experiments on benchmark datasets also demonstrate the superiority of the proposed method.

**Reasons To Reject:**

(1)	First of all, the authors should describe the background of the paper and motivation of this paper in detail. In essence, there are many deep learning-based methods [1-2] which leverages deep learning to solve the zero-shot cross-domain slot filling or recognition problem, authors should analyze their short shortcomings more clearly, and how do these shortcomings promote the creation of the new method.
(2) The figure 2 is ambiguous, and the authors should explain more details about the individual one and its impact on output production.
(3) It would be better to add the Pseudo code of proposed method if possible.

[1] Shah D J, Gupta R, Fayazi A A, et al. Robust zero-shot cross-domain slot filling with example values[J]. arXiv preprint arXiv:1906.06870, 2019.
[2] Shi B, Wang L, Yu Z, et al. Zero-shot learning for skeleton-based classroom action recognition[C]//2021 International Symposium on Computer Science and Intelligent Controls (ISCSIC). IEEE, 2021: 82-86.

**Reproducibility:**

3: Could reproduce the results with some difficulty. The settings of parameters are underspecified or subjectively determined; the training/evaluation data are not widely available.

**Reviewer Confidence:**

2: Willing to defend my evaluation, but it is fairly likely that I missed some details, didn't understand some central points, or can't be sure about the novelty of the work.

---

> ### Author Rebuttal · Authors · 2023-08-29
>
> Thank you for the detailed reviews.
>
> *R1: “describe the background of the paper and motivation of this paper in detail”, “ analyze their short shortcomings more clearly”*
>
> The existing competitive metric-based methods mainly have the following shortcomings:
> + Most of them adopt the two-pass pipeline fashion (i.e., train boundary and typing modules separately), which motivates us to design an end-to-end framework for efficient inference and knowledge sharing between sub-modules.
> + Most of them tend to utilize discrete and context-free label embeddings (i.e., solely encode each label name and lack of interaction with the utterance context), which motivates us to utilize the context-aware soft label embeddings with rich semantics as the supervision information.
> + Regarding domain adaptation capacity, instance-level contrastive learning has been widely explored before, which is sub-optimal to slot filling as a fine-grained labeling task. Considering slot entities tend to embody domain-specific nature in contrast to the counterpart context, we investigate the slot token level contrastive learning scheme to learn the discriminative representations of slot entities for domain adaptation.
>
> Mainly inspired by the above several key points, we ultimately propose our improved method. We will clarify our motivation more clearly in our introduction and related work sections in the final revision.
>
> *R2: “The figure 2 is ambiguous...”*
>
> Thanks for pointing out this. We will refine the Figure 2 to well group the key components and give more detailed descriptions of the figure. All these will be updated in final version.
>
> *R3: “add the Pseudo code of proposed method”*
>
> Thanks for your suggestion, we will complement the pseudo code to help better understand the implementation ideas in the final version.

---

### Official Review · Reviewer_MsTt · 2023-07-31

**Typos Grammar Style And Presentation Improvements:** Please refer to the writing issues in…
**Soundness:** 4

**Excitement:**

4: Strong: This paper deepens the understanding of some phenomenon or lowers the barriers to an existing research direction.

**Missing References:**

None.

**Paper Topic And Main Contributions:**

The main contributions of this paper can be summarized as follows: (1) Compared with existing metric-based methods, this work propose a more efficient and effective end-to-end scheme for zero-shot slot filling, and show the proposed soft label embeddings perform much better than previous commonly-used static label representations. (2) This work investigate the slot-level contrastive learning to effectively improve generalization capacity for zero-shot slot filling.

**Questions For The Authors:**

I hope the author can response to the experiment problems. If the author can provide more results on other datasets or give a clear explaination about using only SNIPS dataset, I would be inclined towards improving the **soundness** score.

**Reasons To Accept:**

(1) The proposed end-to-end scheme for zero-shot slot filling demonstrates a notable advance over conventional two-stage methods, showcasing enhanced simplicity, elegance, and notably improved efficiency. The zero-shot performance and the inference speed are both satisfactory.

(2) In addition to the overall end-to-end pipeline, the author also propose a soft context-aware label embedding and a slot-level contrastive learning, to further improve the zero-shot performance.

(3) The experiments is extensive and verify the effectiveness of the approach.

**Reasons To Reject:**

For **experiment**, I notice LEONA (Siddique et al. 2021) also adopt other datasets (e.g. ATIS, SGD) to evaluate the slot-fliing task. So I recommend the author to provide more results on these datasets.

Also, I believe there are some minor **writting** suggestions I can offer.

(1) The alignment between the module names in Figure 1 and the corresponding titles in the main text appears to be less than optimal. For instance, 'Context-aware Label Embedding' and 'Slot Boundary Detection' do not possess direct visual linkages within the model framework. Such misalignment could potentially impede readers, especially those who are less acquainted with the task, and necessitate additional efforts to locate pertinent information.

(2) Source slot type number can be added in line121 since you specify the target slot type number.

**Reproducibility:**

4: Could mostly reproduce the results, but there may be some variation because of sample variance or minor variations in their interpretation of the protocol or method.

**Reviewer Confidence:**

3: Pretty sure, but there's a chance I missed something. Although I have a good feel for this area in general, I did not carefully check the paper's details, e.g., the math, experimental design, or novelty.

---

> ### Author Rebuttal · Authors · 2023-08-29
>
> Thank you for your detailed reviews.
>
> *R1: "provide more results on other datasets"*
>
> We conduct experiments with the cross-dataset setting where our model is trained on one dataset and evaluated on another one. Due to deadline time limitation, we just verify our method on SNIPS (multi-domain) and ATIS (single-domain). The results (F1) are shown as follows:
>
> | **Model**            | **SNIPS->ATIS** | **ATIS->SNIPS** |
> |---------------------|-----------------|-----------------|
> | Coach               | 14.35           | 9.76            |
> | LEONA               | 20.80           | 14.36           |
> | Ours | 25.40           | 14.49           |
>
> The results demonstrate our method can still work in the challenging cross-dataset setting.
>
> We will complement the complete experiments of cross-dataset settings (including large-scale multi-domain dataset SGD) and provide detailed analysis in our final version.
>
> *R2: “some minor writing suggestions”*
>
> Thanks for pointing out these. We will clear these writing issues and update in the final version.

---

### Official Review · Reviewer_aDbx · 2023-08-04

**Soundness:** 4

**Excitement:**

3: Ambivalent: It has merits (e.g., it reports state-of-the-art results, the idea is nice), but there are key weaknesses (e.g., it describes incremental work), and it can significantly benefit from another round of revision. However, I won't object to accepting it if my co-reviewers champion it.

**Paper Topic And Main Contributions:**

This paper focuses on the task of slot filling and its recent developments using deep learning and large-scale annotated data. The authors address the critical challenge of handling novel domains in slot filling, where samples from these domains were not seen during training, leading to degraded recognition performance due to domain shifts. The main contributions of the paper lie in the design of a more efficient and effective end-to-end metric learning scheme for zero-shot slot filling, the introduction of context-aware soft label representations, and the exploration of slot-level contrastive learning to enhance generalization capacity. The experimental evaluation further validates the superiority of the proposed approach and contributes valuable insights into the task of slot filling.

**Reasons To Accept:**

1) The paper proposes a novel end-to-end metric learning scheme specifically tailored for zero-shot slot filling, which is more efficient and effective compared to existing metric-based methods. This new approach offers a fresh perspective and contributes to advancing the field of slot filling.
2) The introduction of context-aware soft label representations presents a significant improvement over commonly-used static label representations. This innovation showcases the paper's ability to enhance the performance of the model through novel representation techniques.
3) The investigation and incorporation of slot-level contrastive learning demonstrate the paper's comprehensive exploration of different learning strategies to effectively improve the generalization capacity for zero-shot slot filling.

**Reasons To Reject:**

1) While the paper introduces context-aware soft label representations and shows their superiority over commonly-used static label representations, it lacks a comprehensive comparison with other existing soft label embedding methods. A more thorough analysis of various soft label embedding techniques would enhance the credibility of the proposed approach.
2) The comparison in Table 1 is made against relatively outdated baseline methods. To provide a more meaningful evaluation, the paper should consider including state-of-the-art methods that have emerged since the development of the paper. This would help in understanding how the proposed approach performs in comparison to the latest advancements in the field.
3) The paper introduces a loss function in Equation (9) with three terms, but lacks a thorough explanation for choosing a 1:1:1 weighting for these terms. The justification for this specific choice is not adequately provided, raising questions about the rationale behind the loss function design.

**Reproducibility:**

4: Could mostly reproduce the results, but there may be some variation because of sample variance or minor variations in their interpretation of the protocol or method.

**Reviewer Confidence:**

2: Willing to defend my evaluation, but it is fairly likely that I missed some details, didn't understand some central points, or can't be sure about the novelty of the work.

---

> ### Author Rebuttal · Authors · 2023-08-29
>
> Thank you for the thoughtful and detailed reviews.
>
> *R1: “comprehensive comparison with other existing soft label embedding methods”*
>
> The soft label embeddings in this work are closely derived from the framework design (e.g., need interact with the utterance context representations). Since they are not isolated components, changing to other existing soft label embedding scheme may not fit our framework well and need to adjust the overall structure, which may lead to unfair comparisons among different soft label embedding methods. There may be more powerful and suitable soft label embeddings to further improve our current  model, we are very willing to explore that in depth. Thanks for your insightful suggestions.
>
>
> *R2: “consider including state-of-the-art methods that have emerged”*
>
> Thanks for your suggestion. We will complement the latest advancements (e.g., prompt-based generative baseline methods) to provide a more comprehensive evaluation in the final version.
>
>
> *R3: “lacks a thorough explanation for choosing a 1:1:1 weighting for these terms”*
>
> As a cascade-style framework, the impact of each module is closely correlated. The accuracy of slot boundary prediction directly affects the effect of type prediction, and meanwhile the accuracy of type prediction determines the final recognition performance. Thus, these two parts (i.e., L_{bdy} and L_{typ}) of training objective are considered equally important. L_{ctr}, though as an auxiliary loss, plays a key role in improving domain adaptation capability. Our preliminary ablation experiments (not presented in the original paper) also showed the loss weights of 1:1:1 in Eq (9) could yield relatively better performance (as listed in the following table):
> | **$L_{bdy}$: $L_{typ}$: $L_{ctr}$** | **F1** |
> |-----------------------|:------------------:|
> | 1: 1: 1                   | 49.67              |
> | 1: 0.5: 0.5               | 46.42              |
> | 1: 0.5: 0.1               | 45.34              |
> | 1: 0.1: 0.5               | 41.30              |
> | 1: 1: 0.5                 | 47.72              |
>
> We will complement more detailed explanation about the loss function design in the final version.

---

### Meta-Review · Area_Chair_YJ2z · 2023-09-19

**Recommendation:** 5

**Metareview:**

This paper tackles zero-shot cross-domain slot filling, a challenging but meaningful setting. It proposes a novel adaptive end-to-end metric learning method that includes context-aware soft label representations and slot-level contrastive learning. Experiment results on the SNIPS dataset as well as under cross-dataset settings (from rebuttal) show competitive performance of the proposed method.

Strengths:
- Innovative, well-motivated method designs
- Clear and stylish writing
- Competitive empirical results

Weaknesses:
- As the reviewers pointed out, the clarity in several places, including motivation, could still be improved. the authors are strongly encouraged to revise accordingly in the future revision.

---

### Decision · Program_Chairs · 2023-10-07

**Decision:**

Accept-Main

**Comment:**

This paper tackles zero-shot cross-domain slot filling, a challenging but meaningful setting. It proposes a novel adaptive end-to-end metric learning method that includes context-aware soft label representations and slot-level contrastive learning. Experiment results on the SNIPS dataset as well as under cross-dataset settings (from rebuttal) show competitive performance of the proposed method.

Strengths:
- Innovative, well-motivated method designs
- Clear and stylish writing
- Competitive empirical results

Weaknesses:
- As the reviewers pointed out, the clarity in several places, including motivation, could still be improved. the authors are strongly encouraged to revise accordingly in the future revision.